# High Accuracy, Less Talk (HALT): Reliable LLMs via Capability-Aligned Finetuning

**Tim Franzmeyer**[1][†]    **Archie Sravankumar**[2][†]    **Lijuan Liu**[3]    **Yuning Mao**[3]    **Rui Hou**[3]

**Sinong Wang**[3]    **Jakob Foerster**[1,3]    **Luke Zettlemoyer**[3,4]    **Madian Khabsa**[3]

[1]University of Oxford    [2]Anthropic    [3]Meta    [4]University of Washington

[†]Work done while at Meta.

## Abstract

Large Language Models (LLMs) currently respond to every prompt. However, they can produce incorrect answers when they lack knowledge or capability – a problem known as hallucination. We instead propose post-training an LLM to generate content only when confident in its correctness and to otherwise (partially) abstain. Specifically, our method, HALT, produces capability-aligned post-training data that encodes what the model can and cannot reliably generate. We generate this data by splitting responses of the *pretrained LLM* into factual fragments (atomic statements or reasoning steps), and use ground truth information to identify incorrect fragments. We achieve capability-aligned finetuning responses by either removing incorrect fragments or replacing them with "Unsure from Here" – according to a tunable threshold that allows practitioners to trade off response completeness and mean correctness of the response's fragments. We finetune four open-source models for biography writing, mathematics, coding, and medicine with HALT for three different trade-off thresholds. HALT effectively trades off response completeness for correctness, increasing the mean correctness of response fragments by 15% on average. By tuning HALT for highest correctness, we train a single reliable Llama3-70B model with correctness increased from 51% to 87% across all four domains while maintaining 53% of the response completeness achieved with standard finetuning.

## 1 Introduction

Most current language models attempt to respond to the majority of prompts, regardless of how complex they are or how much domain knowledge is required to answer them. While this behavior is desirable for creative tasks (e.g., poem writing), it is undesired when factual correctness is crucial. This phenomenon, often referred to as hallucination, poses risks in high-stakes applications such as medicine or law. We propose HALT, which finetunes an LLM to only respond with information it is confident about, and to (partially) abstain otherwise. For example, when asked to write a person's biography, the LLM would list only facts it is confident in and would omit others. Similarly, when presented with a complex math problem, it would show only the reasoning steps in which it has high confidence, terminating its response with "Unsure from here" if it cannot proceed – potentially forgoing a final answer. Naturally, restricting an LLM to provide only high-confidence information reduces the total number of correct statements, as it withholds any information it deems uncertain – even if some of that information might ultimately be correct. To balance this trade-off, HALT allows practitioners to adjust a confidence threshold. This threshold controls whether the LLM should respond more eagerly (and risk occasional errors) or more conservatively (and risk omitting correct information).

We remark that no model will ever achieve perfect performance across all possible tasks, requiring capability-aligned responses regardless of model strength (OpenAI, 2025). Additionally, HALT is not about distilling from a stronger model; rather, it directly addresses the problem of ensuring that an LLM's outputs are aligned with its own uncertainty.

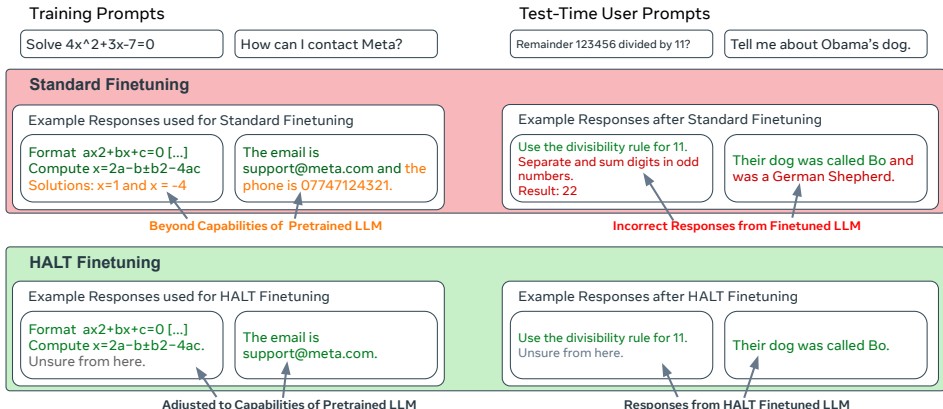

Figure 1: Comparison between Standard and HALT Finetuning for LLMs: Standard finetuning trains the LLM on responses that potentially exceed the pretrained LLM's capabilities, which results in incorrect outputs at test time. HALT finetuning trains the model only on content within the pretrained LLM's capability limits, omitting unknowns or replacing them with "Unsure from here". HALT finetuning improves response correctness as the LLM is trained to generate responses according to its capabilities, which may result in partially incomplete responses when the pretrained LLM's capabilities are insufficient.

HALT's key insight is to finetune the pretrained LLM only on *correct content it is capable of generating*, i.e., content within the bounds of the knowledge and reasoning capabilities obtained during pretraining, as we illustrate in Figure 1. Further, HALT finetunes the LLM to output "Unsure from here" when uncertain about the remainder of the generation.

HALT is motivated by the observation that LLMs possess internal calibration (Tian et al., 2023b; Kadavath et al., 2022), suggesting they can produce responses aligned with their internal confidence in the relevant facts or reasoning steps. HALT's approach of generating capability-aligned finetuning samples according to the pretrained LLM's capabilities is based on recent work (Lin et al., 2023; Zhou et al., 2024) which shows that LLMs do not acquire novel capabilities during finetuning but only learn to effectively utilize knowledge and reasoning capabilities obtained during pretraining. Moreover, HALT's approach of finetuning only on content that the LLM is capable of generating is supported by recent work (Gekhman et al., 2024; Kang et al., 2024) showing that finetuning LLMs on concepts unknown to them increases hallucination. While HALT requires additional model generations and post-processing for each pretrained LLM it is applied to, we believe this is reasonable, as recent work (Zhou et al., 2024) found that merely 1000 finetuning examples can be sufficient to finetune generally capable models. Unlike previous methods (Yadkori et al., 2024; Farquhar et al., 2024; Cheng et al., 2024; Brahman et al., 2024; Feng et al., 2024; Zhang et al., 2024a; Kang et al., 2024), HALT requires no additional computations at test time, as it does not rely on post-processing or sampling. Instead, it produces a finetuned model that attempts the best possible answer within its capability. Also, unlike earlier work, HALT extends to prompts involving not only knowledge retrieval but also complex reasoning.

Given a pretrained LLM and a finetuning dataset, for each prompt, HALT generates a response aligned to the capabilities of the LLM. For a given prompt, the HALT pipeline (1) generates an initial response via few-shot prompting of the pretrained LLM, (2) splits it into *factual fragments*, (3) assesses correctness of individual fragments via an *Evaluator LLM* with access to ground truth information (e.g. the ground truth response), and (4) post-processes the fragments to arrive at the final HALT finetuning response – which only contains fragments that the pretrained LLM is capable of generating. Figure 2 provides an example of the pipeline.

For the second step of decomposing the response into fragments, we make the assumption that responses either consist of *independent fragments* or *causally dependent fragments*, i.e., fragments that build on top of each other in a logical sequence. This distinction is made for each prompt based on its domain, e.g., math samples are assumed to consist of causally dependent fragments, while biographies are assumed to consist of independent ones.

For responses composed of *independent* fragments, we rely on prior work (Song et al., 2024) and use a finetuned LLM to extract atomic statements, i.e., independently verifiable statements. For example, "Barack Obama was born in Hawaii" is an atomic statement that could be extracted from his biography. In step 3, the correctness of each fragment is assessed independently using the ground-truth-conditioned evaluator LLM (e.g. conditioned on the respective Wikipedia article). The HALT finetuning response (4) is then composed of all correct fragments, or set to "Unsure from here" if no fragments were correct.

For responses composed of causally *dependent* fragments, such as responses to math questions, HALT instead splits responses at natural boundaries like new lines or equation statements. HALT then determines the fragment at which the first error occurs using an evaluator LLM with access to the ground truth response. The HALT response is then composed by replacing the incorrect fragment and all subsequent fragments with a single "Unsure from here" statement, arriving at a capability-aligned finetuning response.

Importantly, HALT enables practitioners to choose the desired trade-off between response *completeness* (analogous to *Recall*) and response *correctness* (analogous to *Precision*) according to the deployment scenario. This is effected by estimating model capabilities based on multiple responses of the pretrained LLM. For example, composing the HALT response from a response with less-than-average correct fragments yields a conservative estimate of the model's capability, resulting in an LLM that responds more carefully.

We evaluate HALT on a wide variety of language models, including LLama3-8B, LLama3-70B, Gemma2-9B, and Mistral-7B, and on a series of tasks, including writing Wikipedia-style biographies, solving competition mathematics problems, answering medical questions, and solving coding problems posed in natural language. We compare HALT to standard finetuning, prior work on factuality improvement (FactTune, (Tian et al., 2023a)), prior work on training models to abstain (IDK, (Cheng et al., 2024)), and a heuristic baseline that trims responses to match the response length of HALT. We split generated responses from prompts in the eval dataset into factual fragments and assess the completeness and average correctness of fragments for each response. Tuning HALT based on our threshold allows to increase the correctness by 17% for Llama3-70B on average. Across all settings, HALT improves the F1 score, i.e. the harmonic mean of completeness and correctness, on average by 4%, as compared to the baselines. Tuning HALT for correctness, we combine data from all domains and finetune a single *reliable* LLama3-70B model that achieves an accuracy of 87% across the four domains – which is in contrast to 51% accuracy achieved with standard finetuning – while maintaining 51% completeness of responses. Last, we demonstrate that slightly modifying HALT to annotate fragments as "Unsure" – instead of omitting them – informs users about uncertainty while retaining complete responses.

## 2 METHODS

**Overview.** A pretrained LLM $\mathcal{M}$ maps from input sequences to output sequences, $\mathcal{M} : \mathcal{X} \rightarrow \mathcal{Y}$. Additionally, we have a finetuning dataset $D = (x_j, y_j)_{j=1}^m$, which consists of $m$ pairs of prompts $x_j$ and their corresponding ground truth responses $y_j$. The goal of our method, which we refer to as HALT, is to create a modified finetuning dataset $D^H = (x_j, y_j^H)_{j=1}^m$, where each target response $y_j^H$ is aligned with the capabilities of the pretrained model $\mathcal{M}$. We assume access to an evaluator $\mathcal{E}$ that takes as input a statement $f$ and information $J$ and determines whether $f$ is correct or incorrect according to the given information, i.e., $\mathcal{E} : (f, J) \rightarrow \{0, 1\}$. This evaluation could be performed by a human annotator, an additional LLM, or any other means.

Figure 2 provides an overview of how HALT processes each prompt $x_j$: First, a preliminary response is generated using the pretrained model $\mathcal{M}$, then split into factual fragments. Next, the evaluator $\mathcal{E}$ assesses the correctness of individual fragments, using the given ground truth response $y_j$, potentially supplemented by other relevant context. Finally, incorrect fragments are removed or replaced by "Unsure from here" to arrive at the target response $y_j^H$.

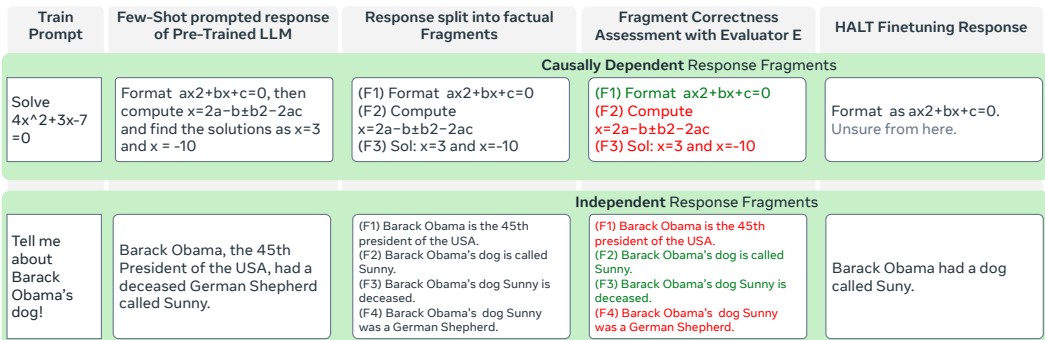

Figure 2: The HALT pipeline includes generating a preliminary response via few-shot prompting of the pretrained LLM, splitting it into factual fragments, and assessing each fragment's correctness before compiling the HALT finetuning response. For a causally dependent response, as displayed at the top, an error results in an "Unsure from here" marker to indicate uncertainty. For a response consisting of independent fragments, as shown at the bottom, incorrect fragments are removed.

## 2.1 CREATING A CAPABILITY-ALIGNED FINETUNING DATASET WITH HALT

**Generating the Preliminary Response.** For each prompt $x_j$, we first generate a response that aligns with the format of the ground truth responses in the finetuning dataset $D$ through few-shot prompting of the pretrained model. We sample a set of four prompt-response pairs $\{(x_k, y_k)\}_{k=1}^4$ uniformly at random from $D \setminus \{(x_j, y_j)\}$ as in-context learning examples. The context $C_j = \{(x_k, y_k)\}_{k=1}^4$ is concatenated with the target prompt $x_j$ using a question-answer format. The pretrained model $\mathcal{M}$ is then prompted to create a preliminary response $y_j^{\text{pt}}$ for the prompt $x_j$, i.e. $y_j^{\text{pt}} = \mathcal{M}(\text{concat}(C_j, x_j))$, where concat denotes the concatenation of the context $C_j$ and the target prompt $x_j$ into a single input sequence.

**Splitting the Preliminary Response into Factual Fragments.** Next, we split the preliminary responses $y_j^{\text{pt}}$ into individual factual fragments, denoted as $y_j^{\text{pt}} = (f_1, \ldots, f_k)$, where each fragment represents a verifiable unit. The method used for fragmentation generally depends on the type of response. Simple, inherently structured responses may allow fragmentation along natural boundaries, such as equal signs in math equations, bullet points or newline characters. Long and complex sentences may require more sophisticated methods for fragmentation, such as machine learning methods specifically trained to extract verifiable statements from complex sentences (Song et al., 2024; Min et al., 2023; Wei et al., 2024). We empirically find that grouping prompts by type of question (e.g., math, coding, knowledge) allows us to determine the required fragmentation method. We describe fragmentation in Section 3.1.

**Assessing Correctness of Fragments.** We distinguish between two types of response structures when determining the correctness of the individual response fragments $f_1, \ldots, f_k$: Responses consisting of either *independent fragments*, or of *causally dependent fragments* that build on each other in a logical sequence. Figure 2 shows the HALT processing pipeline for example prompts of both categories. We note that not every response will perfectly fit one of the two categories but leave more complex dependency structures, such as dependency graphs, for future work.

For responses consisting of independent fragments, the correctness of each fragment is independent of the correctness of other fragments. Examples are Wikipedia-style information overviews, lists of independent recommendations, and results for independent tasks. For responses consisting of causally dependent fragments, the correctness of a fragment depends on the correctness of prior fragments. Examples are step-by-step solutions, code implementations, or sequential reasoning, where each part builds upon previous ones.

For responses consisting of independent fragments, we assess the correctness of each fragment individually using the evaluator $\mathcal{E}$, with information $J$ given by the ground truth response $y_j$ and potentially additional information. For responses consisting of causally dependent fragments, we

prompt the evaluator $\mathcal{E}$ – conditioned on the ground truth response – to identify the first incorrect fragment of the response, and consider all subsequent fragments as incorrect.

**Creating the** HALT **Finetuning Response.** For responses with independent fragments, we merely remove any incorrect fragments. For responses with causal dependency, we remove all statements starting from the first incorrect statement, and, in case any incorrect statements were present, place "Unsure from here" at the end of the response, as illustrated in Figure 2.

## 2.2 TRADING OFF BETWEEN RESPONSE CORRECTNESS AND COMPLETENESS

The trade-off between average correctness of fragments and response completeness can be tuned by choosing the preliminary response from a set of sampled responses of the pretrained LLM. Specifically, instead of relying on a single greedily decoded response to estimate the pretrained model's capabilities, we sample multiple preliminary responses. Selecting the "worst" response among the sampled preliminary responses, i.e., the response in which the lowest number of fragments is correct, provides a conservative estimate of the model's capability. Composing the HALT response based on the worst response results in a finetuned model with higher correctness but lower completeness. The opposite holds true when selecting the best response instead.

Specifically, for each prompt $x_j$, we sample $N$ preliminary responses $\{y_j^{\text{pt},n}\}_{n=1}^N$ with different random seeds and compute the average fragment correctness for each response as outlined in Section 3.1. We sort the preliminary responses in ascending order of average correctness and select the $\alpha$-percentile response as $n^* = \lceil \alpha N \rceil$ and $y_j^{\text{pt}} = y_j^{\text{pt},n^*}$. Here, $\alpha$ determines the trade-off, where a lower $\alpha$ favors correctness over completeness. The selected response $y_j^{\text{pt}}$ is then processed with the HALT pipeline shown in Figure 2.

## 3 EXPERIMENTAL VALIDATION

**Overview.** As HALT responses are derived from few-shot prompted responses of the pre-trained model, we first validate that finetuning on these yields performance similar to finetuning on the ground truth responses. We then assess the accuracy of our evaluator $\mathcal{E}$, which we implement using Llama3-405B. We observe in Table 1 that the evaluator has an average per-response absolute error ranging from 0.27 to 1.14 fragments. We then evaluate HALT on four different datasets and for four open-source LLMs, namely Llama3-8B (Dubey et al., 2024), Llama3-70B (Dubey et al., 2024), Gemma2-9B (Team et al., 2024), and Mistral-7B (Jiang et al., 2023). We demonstrate that HALT allows to trade off between response completeness and correctness while simultaneously achieving the best arithmetic mean of completeness and correctness. Utilizing HALT, we train a single reliable model on all four datasets that achieves 87% correctness, an increase of 36% compared to the correctness achieved by standard finetuning. Last, we demonstrate that HALT can be modified to annotate response fragments that the model is unsure about as "Unsure" – instead of omitting them. This allows to provide users with complete responses, while simultaneously informing users about likely incorrect parts of the response.

## 3.1 IMPLEMENTATION AND VALIDATION OF RELEVANT COMPONENTS

**Prior Work and Baselines.** We implement FactTune (Tian et al., 2023a), IDK (Cheng et al., 2024), and supervised finetuning, which we refer to as "Unchanged". We remark that none of these baselines allow for adjusting the trade-off between completeness and correctness. FactTune, applicable to knowledge tasks, i.e. the Wikipedia dataset, first finetunes the pretrained model, e.g. on Wikipedia biographies. It then generates responses from the finetuned model, evaluates atomic statements using FactScore (Min et al., 2023), and creates a preference dataset to finetune the model again using Direct Preference Optimisation (Rafailov et al., 2024). IDK evaluates which prompts a model can answer, then finetunes the model to either entirely abstain or completely answer to given prompts. We additionally implement a *RandomTrim* baseline, which, instead of removing fragments according to the capability of the pretrained model as done with HALT, removes the $n$ last fragments of a given response. $n$ is sampled from a Poisson distribution, which is chosen such that the average response length matches that of HALT responses.

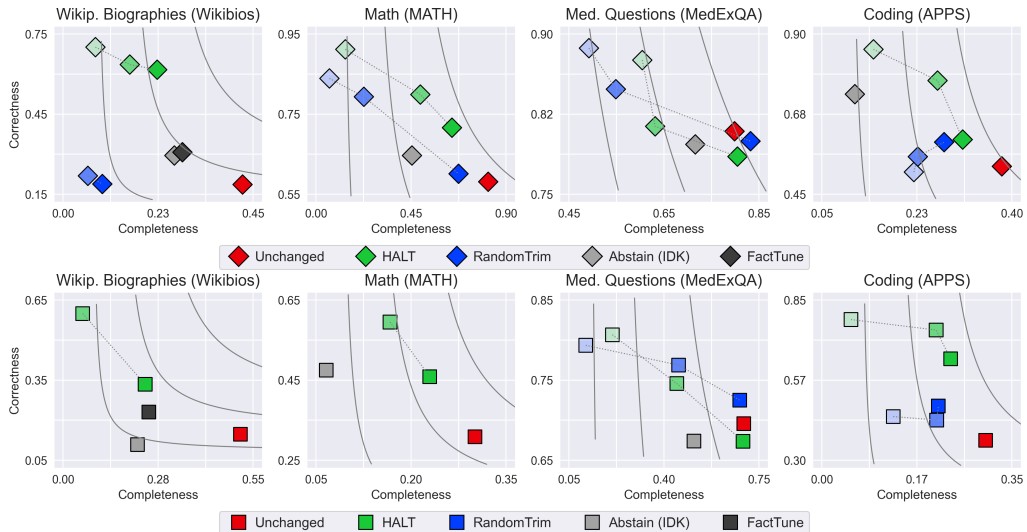

Figure 3: Response correctness (y-axis) and completeness (x-axis) for LLama3-70B (top) and Llama3-8B (bottom) when finetuned with different methods (desired number of fragments is $n_{all}$). F1 Score is constant along curved grey lines, and highest in the top right corner. For HALT and Randomtrim, results are shown with different trade-offs between completeness and correctness, where lighter colors indicate tuning for higher correctness. We omit results with less than 5% completeness, e.g., RandomTrim results for Wikipedia Biographies and Math. We observe that HALT allows for strongly influencing the trade-off between correctness and completeness across all four datasets while achieving higher F1 scores (closer to the top right corner) than baseline methods. We show results for Gemma2-9B and Mistral7B in Figure 7 in the Appendix.

**Datasets.** Wikibios (Lebret et al., 2016) contains the *summary paragraph* of individuals listed on Wikipedia, sampled uniformly randomly. Hence, this dataset contains many individuals who are only little known. Since the original dataset was assembled 8 years ago, we updated all information using the official Wikipedia API, and excluded all individuals not uniquely identified when searching for their name on the Wikipedia API. We added each person's full Wikipedia article to the dataset, which is used as additional information for the evaluator $\mathcal{E}$. In the reasoning domain, we only consider datasets that provide step-by-step solutions, as we found that these increase the accuracy of the evaluator $\mathcal{E}$ significantly. MATH (Hendrycks et al., 2021b) contains mathematical problems across various fields. MedExQA (Kim et al., 2024) is a medical question-answering dataset covering five distinct specialties. APPS (Hendrycks et al., 2021a) evaluates code generation, containing a wide range of programming challenges; we consider those marked as "introductory".

**Splitting Responses into Fragments.** We decompose a response into factual fragments as follows. Wikipedia answers are often complex and nested, which prohibits a heuristic-based fragmentation. We instead use a state-of-the-art fact-extraction LLM, made available by Song et al. (2024). Although responses are well structured for MATH, we found that a heuristic-based fragmentation yields unsatisfactory results and instead prompted LLama3-405B to fragment responses at natural boundaries. For MedExQA we found it satisfactory to split answers at full stop signs. Similarly, code responses in APPS are split line-for-line.

**Evaluator Implementation.** For Wikibios, we prompt Llama3-405B with the fragment to assess, as well as with the entire Wikipedia article of the relevant person, and ask it to assess whether the given fragment is correct or incorrect in the given context (see example prompt in App. B.1). Hence, the evaluator is called multiple times to assess a single response. For MATH, MedExQA, and APPS, we prompt Llama3-405B with the numbered fragments that make up the response and the ground-truth step-by-step solution, and prompt it to identify the first incorrect fragment.

**F1 Score to Measure the Mean of Completeness and Correctness.** We follow Wei et al. (2024) and use the F1 score as a combined measure of response correctness and completeness, analogous to

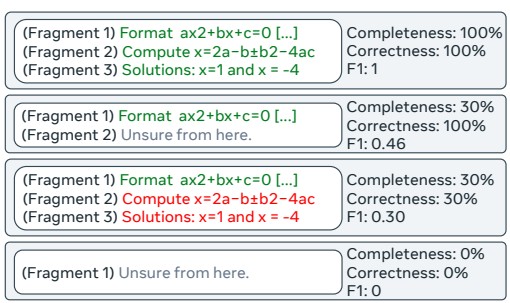

Figure 4: Example responses (ground truth response in the top left corner), with completeness, correctness, and F1 scores annotated for each. The number of required fragments here is three. We observe that the highest F1 score is achieved when the LLM answers with the correct first fragments and abstains after.

Table 1: Avg. number of fragments per response and avg. number of fragments incorrectly labelled by the Llama3-405B evaluator. Analysis run on 300 manually-labelled Llama3-70B responses per dataset.

|  | Wikipedia | MATH | Medical Q. | Coding |
|---|---|---|---|---|
| Avg # fragm. | 9.58 | 5.54 | 5.26 | 7.85 |
| Avg # incorrect | 0.27 | 0.63 | 0.41 | 1.14 |

Table 2: Finetuning models on responses generated via best-of-5 few-shot prompting results in only marginally lower performance (average correctness of statements) than finetuning on the *Unchanged* (ground truth) responses.

| Pretrained Model | Responses | MATH | Medical Q. | Coding |
|---|---|---|---|---|
| Llama-3-8B | Unchanged | 0.30 | 0.69 | 0.30 |
|  | Few-Shot | 0.25 | 0.68 | 0.27 |
| Llama-3-70B | Unchanged | 0.81 | 0.80 | 0.38 |
|  | Few-Shot | 0.76 | 0.78 | 0.35 |
| Mistral-7B | Unchanged | 0.30 | 0.69 | 0.30 |
|  | Few-Shot | 0.25 | 0.69 | 0.24 |
| Gemma-2-9B | Unchanged | 0.39 | 0.69 | 0.27 |
|  | Few-Shot | 0.33 | 0.71 | 0.26 |

binary classification. Here, response completeness (Recall) measures the ratio of correct fragments relative to the desired number of fragments, i.e., $\frac{n_{\text{correct}}}{n_{\text{desired}}}$. Response correctness (Precision) is the relative correctness of all given response fragments, i.e., $\frac{n_{\text{correct}}}{n_{\text{given}}}$. We consider two definitions for the desired number of fragments. The first defines the desired number of fragments as the number required to answer the question fully (see App. A.1 for details), which we refer to as $n_{\text{all}}$ (which is independent of the model's capability). Hence, any method can only achieve full recall under this definition if the pretrained LLM's capabilities are sufficient to answer all questions completely and correctly. The second option defines the number of desired fragments according to the pretrained LLM's capability, which we refer to as $n_{\text{capable}}$. Specifically, $n_{\text{capable}}$ is the number of *correct* fragments in the few-shot prompted response of the pretrained LLM. We remark that under the latter definition, a method could attain full recall (and an F1 score of 1) if the finetuned LLM answers exactly with those response fragments that are within its capability limits. Figure 4 shows response completeness and correctness for different example responses, with the number of desired fragments defined as the number required to answer the question fully ($n_{\text{all}}$).

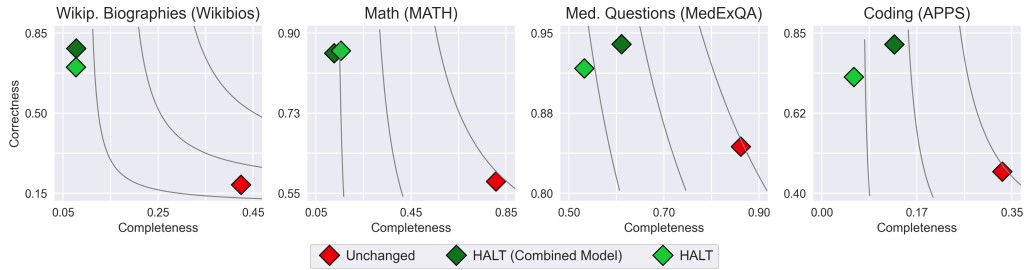

Figure 5: We show results for training a single *reliable* LLama3-70B model with HALT tuned for increased correctness, trained on equal shares of all four datasets, referred to HALT (Combined Model). We additionally plot results for HALT trained on each dataset individually, referred to as HALT, and results when finetuning on the Unchanged dataset. We observe that HALT allows for increasing average accuracy across all datasets by 36% to 87% while maintaining an average completeness of 25%.

**Information-Conditioned Llama-3-405B Yields a Strong Correctness Evaluator.** We evaluate the performance of our evaluator in assessing the correctness of individual fragments (denoted as $f_1, \ldots, f_k$ for a response split into $k$ fragments, as defined in Section 4) of responses of all four

datasets. For each dataset, we randomly sampled 300 responses generated by Llama3-70B and manually annotated the correctness of all fragments. We observe in Table 1 that the Llama3-405B evaluator's error ranges from 0.27 incorrectly assessed atomic statements per Wikipedia response, composed of 9.58 atomic statements on average, to a discrepancy of 1.14 lines on Coding, with an average number of 7.85 lines of code per response. In MATH, the evaluator misjudges the location of the first incorrect fragment by only 0.6 reasoning steps, at an average response length of 5.54. We conclude that Llama3-405B – prompted with relevant information – is well suited to assess the correctness of individual logical fragments.

**Finetuning on Few-Shot Prompted Responses is comparable to Finetuning on Ground Truth Responses.** HALT relies on responses derived from few-shot prompting of the pretrained LLM. For the MATH, Medical QA, and Coding datasets, we compare the performance of the finetuned model when either finetuned on the dataset's unchanged ground truth response or the best-of-5 few-shot prompted response of the pretrained LLM. Table 2 shows that the average gap ranges from 1.7% to 3.7% only across the models, and could be further closed by sampling more responses. As a result, the expected correctness for finetuning methods that rely on the ground truth response is 1.7% to 3.7% higher than that of HALT responses; however, we empirically find that HALT can still outperform such methods. We remark that for Mistral-7B, the gap is at 5% and 6% in Math and Coding, respectively, suggesting that this model has worse in-context learning capabilities than the other examined models. As discussed later, we found that this performance gap influences the finetuned Mistral-7B HALT models. These findings support the assumption that LLMs do not acquire novel knowledge or capabilities during finetuning (Lin et al., 2023; Zhou et al., 2024).

## 3.2 HALT Enables Trading off Completeness and Correctness

Next, we finetune all four models on all four datasets, comparing HALT to the baselines. For HALT, we sample five few-shot prompted preliminary responses, sort them by relative correctness, choose a response according to $\alpha \in [40\%, 60\%, 80\%]$. That is, we either choose the most, second most, or third most accurate response, and then process it with the HALT pipeline. We provide examples of finetuning responses for different tradeoff parameters $\alpha$ in App. B.2. We discard the response with the lowest and second lowest relative correctness, as we empirically found that finetuning on these can result in close to zero completeness and correctness. Figure 3 shows that choosing different $\alpha$ for HALT allows to trade off correctness and completeness effectively, altering correctness by $\pm$ 17% and $\pm$ 12% for Llama3-70B and Llama3-8B, respectively. Table 3 in the Appendix shows that HALT further achieves the highest harmonic mean of completeness and correctness in most cases, as compared to baseline methods, both for $n_{\text{desired}}$ set to $n_{\text{all}}$ and to $n_{\text{capable}}$.

**HALT Does Not Harm General Instruction-Following Capabilities.** To assess whether HALT maintains coherence and general instruction-following capabilities beyond domain-specific accuracy, we conducted an AlpacaEval-style evaluation (Li et al., 2023) comparing Llama3-70B HALT (trained with $\alpha = 0.6$) against the base Llama3-70B model across 578 prompts spanning all four evaluation datasets, using GPT-5 as a judge. HALT achieves near-parity with the base model overall (50.8% win rate), demonstrating that selective abstention does not significantly harm general instruction-following capabilities. Notably, HALT outperforms the base model on wikibios_v2 (66.0% win rate), where factual accuracy is critical—precisely the use case HALT is designed for. On mathematical reasoning and general-domain tasks (MATH, appsintro, medex), HALT shows modest degradation (41.8-45.4% win rates), expected since judges may penalize abstentions when completeness is valued over guaranteed correctness (see Table 5 in the Appendix for full results).

## 3.3 Training a Multi-Domain LLM Achieving 87% Correctness

We now evaluate the real-world use case of training a single model that achieves high correctness across multiple domains. Specifically, we train a single *reliable* Llama3-70B model on an equal-parts mix of data from all four domains with HALT tuned for increased accuracy, i.e., with $\alpha = 40\%$. Figure 5 shows that this model achieves an average accuracy of 87% across all four domains, as compared to 51% achieved by training on a mix of the ground truth datasets. Meanwhile, the response completeness remains at 25% on average. This demonstrates that HALT allows training a generally capable model that users can trust significantly more than models trained with standard finetuning.

### 3.4 Add-On HALT: Marking Fragments as "Uncertain" instead of omitting them.

Users might desire to observe the model's full response while being informed which parts might be incorrect, enabling them to edit uncertain code or verify statements externally. We evaluate this use case by modifying HALT to annotate uncertain response fragments with "Uncertain", instead of omitting them. Specifically, we modify the finetuning dataset for Math, Medical Q., and Coding by marking the first incorrect response fragment and all subsequent fragments as "Uncertain". Note that these fragments are replaced by "Unsure from here" in the default HALT response. We do not investigate the open-ended Wikipedia biography writing task as it allows for generating an unbounded number of "Uncertain" statements. Figure 6 in the Appendix shows that training Llama3-70B at $\alpha = 60\%$ with Add-On HALT results in significant correctness improvements when fragments marked as "Uncertain" are excluded from the correctness evaluation, achieving improvements similar to those of HALT's default implementation. When the "Unsure" markers are ignored Add-On HALT achieves response completeness similar to that achieved when finetuning on the Unchanged responses. In conclusion, a slight modification of HALT enables the retention of response completeness while reliably informing the user about likely incorrect fragments.

## 4 Related Work

A large body of prior work studied detecting LLM generations that likely contain hallucinations, i.e., likely contain incorrect statements. Approaches include training probes (Su et al., 2024), directly inspecting hidden states (Chen et al., 2024), or evaluating semantic entropy of generations (Farquhar et al., 2024). These works are complemented by approaches to mitigate hallucinations via weight-space editing (Zhang et al., 2024b), modified decoding techniques (Chuang et al., 2023), preference training on samples that contain fewer or more hallucinations (Tian et al., 2023a), or post-processing of generated responses (Mohri & Hashimoto, 2024; Gui et al., 2024; Wang et al., 2024). In contrast to these works, HALT addresses the trade off between response completeness and correctness, does not require any post-processing of generations at test-time, and applies to reasoning problems.

Recent works have discovered that finetuning LLMs on unknown examples increases hallucinations (Kang et al., 2024; Gekhman et al., 2024; Tian et al., 2023b), which motivates HALT's approach of only finetuning on samples that are within the LLM's capabilities.

Another line of prior work has found that LLMs are well-calibrated, and found that confidence scores can, for example, be obtained via inspection of logits (Kadavath et al., 2022) or via prompting LLMs to state their confidence (Lin et al., 2022). This provides a basis for recent works that train models to abstain when uncertain (Zhang et al., 2024a; Chen et al., 2023; Yadkori et al., 2024; Wen et al., 2024; Brahman et al., 2024; Feng et al., 2024; Cheng et al., 2024; Tuan et al., 2024). Recent work has explored training models to express verbalized uncertainty through confidence scores (Band et al., 2024) and calibrated hedging (Stengel-Eskin et al., 2024). Unlike prior works that either fully abstain, fully answer, or preserve complete responses with uncertainty qualifications, HALT trains LLMs to selectively compose partial responses according to their capabilities, enabling modification of the desired trade off between completeness and correctness.

## 5 Conclusion

We present HALT, a novel finetuning paradigm that trains models to generate responses according to their internal confidence by omitting response fragments they are uncertain about. HALT training yields models that are up to 37% more accurate than those trained using unchanged ground-truth answers while maintaining high response completeness.

**Limitations.** Our method assumes independent or causally dependent fragments; dependency graphs could improve complex tasks. HALT uses binary capability assessment; fine-grained scaling may help. We train separate models per trade-off; a single adaptive model would improve flexibility. Extending to multi-modal domains is promising.

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

# A   SUPPLEMENTARY EXPERIMENTAL DETAILS AND RESULTS

## A.1   DEFINING THE NUMBER OF FRAGMENTS REQUIRED TO FULLY ANSWER A PROMPT.

We remark that the number of fragments necessary to answer a question cannot be unambiguously determined and, in practice, slightly varies for different pre-trained LLMs. However, when, e.g., an LLM responds with two correct fragments, we must define $n_{\text{all}}$ to be able to compute response completeness. As different pretrained LLMs will answer the same few-shot prompt with slightly different numbers of fragments, as shown in Table 4, we choose $n_{\text{all}}$ according to the pre-trained model's response statistics in the MATH and Medical Question datasets. For biography writing and coding, we found that the number of fragments of the ground truth response defines $n_{\text{all}}$ well.

## A.2   ADDITIONAL IMPLEMENTATION DETAILS

**LLM implementations.**   We implement all LLMs using their open-source implementations given at `https://github.com/huggingface/transformers`.

**Finetuning Details.**   We use supervised finetuning with LoRA (Hu et al., 2021) for compute-efficient training, with a rank of 256 and alpha of 512. Training is performed using the AdamW (Kingma, 2014) optimizer with betas set to $(0.9, 0.95)$, weight decay of $0.01$, and an initial learning rate of $1e-5$. The learning rate schedule follows a warmup fraction of 5%, with a decay to $1e-6$ over 5 epochs. The effective batch size is set to 128.

**Computational Resources.**   Finetuning of the 7-9B LLMs took around 5-6 hours on a single Nvidia A100 GPU. Finetuning of the Llama3-70B model took around 3-4 hours on a node of 8 Nvidia A100 (80GB) GPUs.

**Datasets.**   We downloaded the datasets from the following domains. The Wikibios dataset was downloaded from `https://github.com/DavidGrangier/wikipedia-biography-dataset`; the MATH dataset from `https://github.com/hendrycks/math`, the MedexQA dataset from `https://huggingface.co/datasets/bluesky333/MedExQA`, and the APPS dataset from `https://github.com/hendrycks/apps`.

**Baseline Implementations.**   We implemented FactTune (Tian et al., 2023a) using the authors' implementation given at `https://github.com/kttian/llm_factuality_tuning`. For the IDK baseline (Cheng et al., 2024) we follow the authors' implementation given at `https://github.com/OpenMOSS/Say-I-Dont-Know`.

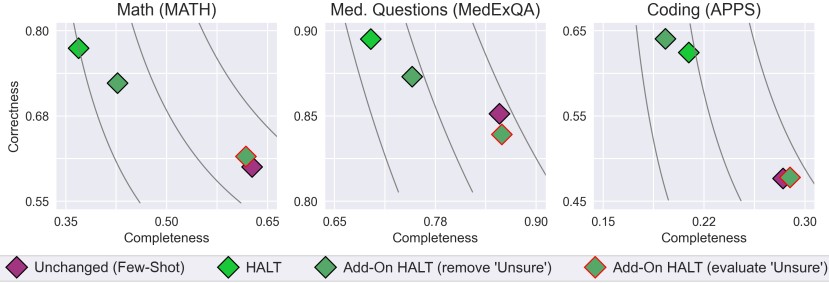

Figure 6: We evaluate an Add-On version of HALT for LLama3-70B. Add-On HALT annotates response fragments as 'Unsure' instead of omitting them. Removing fragments marked by Add-On HALT as 'Unsure' results in a significant increase in correctness. At the same time, evaluating 'Unsure' fragments yields results similar to training on the Unchanged (Few-Shot generated) responses. This demonstrates that Add-On HALT allows for increased correctness while preserving the completeness of unchanged finetuning.

Table 3: F1 score (arithmetic mean of response correctness and completeness) for all models and finetuning methods. We show results when setting the desired number of fragments as those required to fully answer the question (left side of Table) and when setting the desired number as those that the pre-trained model is capable of generating (right side). Note that a perfect F1 score is only attainable when setting $n_{desired} = n_{capable}$. We find that HALT largely outperforms prior work and baselines, by larger margins for $n_{capable}$. For Gemma2-7B, we hypothesize that the relatively lower performance of HALT in Math and coding is likely due to the worse in-context learning capabilities, as outlined in Section 3.1.

| LLM | Finetuning | $n_{desired} = n_{all}$ | | | | $n_{desired} = n_{capable}$ | | | |
|---|---|---|---|---|---|---|---|---|---|
| | | Wikipedia | Math | Medical Q. | Coding | Wikipedia | Math | Medical Q. | Coding |
| Llama-3-8B | Unchanged | 0.23 ± 0.01 | 0.30 ± 0.05 | **0.70** ± 0.02 | 0.33 ± 0.01 | 0.26 ± 0.04 | 0.49 ± 0.07 | 0.81 ± 0.02 | 0.58 ± 0.05 |
| | RandomTrim | 0.05 ± 0.01 | 0.08 ± 0.00 | **0.70** ± 0.02 | 0.30 ± 0.02 | 0.21 ± 0.03 | 0.26 ± 0.00 | 0.82 ± 0.02 | 0.61 ± 0.06 |
| | Abstain (IDK) (Cheng et al., 2024) | 0.14 ± 0.00 | 0.12 ± 0.02 | 0.58 ± 0.16 | 0.05 ± 0.03 | 0.25 ± 0.04 | 0.38 ± 0.10 | 0.69 ± 0.18 | 0.18 ± 0.09 |
| | FactTune (Tian et al., 2023a) | 0.24 ± 0.01 | - | - | - | 0.38 ± 0.02 | - | - | - |
| | HALT (ours) | **0.28** ± 0.00 | **0.35** ± 0.07 | 0.68 ± 0.03 | **0.35** ± 0.04 | **0.63** ± 0.01 | **0.66** ± 0.12 | 0.81 ± 0.04 | **0.77** ± 0.05 |
| Llama-3-70B | Unchanged | 0.26 ± 0.01 | 0.68 ± 0.01 | 0.80 ± 0.03 | **0.44** ± 0.03 | 0.35 ± 0.02 | 0.75 ± 0.01 | 0.87 ± 0.03 | 0.71 ± 0.09 |
| | RandomTrim | 0.12 ± 0.01 | 0.64 ± 0.01 | 0.82 ± 0.00 | 0.38 ± 0.03 | 0.26 ± 0.01 | 0.75 ± 0.01 | 0.88 ± 0.00 | 0.75 ± 0.05 |
| | Abstain (IDK) (Cheng et al., 2024) | 0.28 ± 0.01 | 0.53 ± 0.27 | 0.75 ± 0.07 | 0.19 ± 0.13 | 0.44 ± 0.03 | 0.68 ± 0.38 | 0.82 ± 0.07 | 0.48 ± 0.32 |
| | FactTune (Tian et al., 2023a) | - | - | - | - | - | - | - | - |
| | HALT (ours) | **0.33** ± 0.03 | **0.68** ± 0.04 | **0.81** ± 0.06 | 0.41 ± 0.01 | **0.68** ± 0.02 | **0.87** ± 0.02 | **0.88** ± 0.06 | **0.83** ± 0.04 |
| Mistral-7B | Unchanged | 0.14 ± 0.00 | **0.31** ± 0.02 | **0.69** ± 0.06 | 0.33 ± 0.02 | 0.20 ± 0.04 | 0.50 ± 0.06 | 0.81 ± 0.06 | 0.57 ± 0.05 |
| | RandomTrim | 0.05 ± 0.00 | 0.12 ± 0.06 | 0.64 ± 0.04 | 0.25 ± 0.06 | 0.34 ± 0.04 | 0.43 ± 0.25 | 0.80 ± 0.05 | 0.52 ± 0.37 |
| | Abstain (IDK) (Cheng et al., 2024) | 0.11 ± 0.00 | 0.09 ± 0.02 | 0.56 ± 0.17 | 0.01 ± 0.00 | 0.18 ± 0.03 | 0.43 ± 0.07 | 0.73 ± 0.20 | 0.04 ± 0.02 |
| | FactTune (Tian et al., 2023a) | **0.20** ± 0.01 | - | - | - | 0.44 ± 0.03 | - | - | - |
| | HALT (ours) | 0.13 ± 0.01 | 0.28 ± 0.03 | 0.66 ± 0.02 | **0.34** ± 0.03 | **0.68** ± 0.05 | **0.69** ± 0.41 | 0.81 ± 0.08 | **0.82** ± 0.06 |
| Gemma-2-9B | Unchanged | **0.15** ± 0.01 | 0.42 ± 0.08 | 0.71 ± 0.01 | 0.34 ± 0.02 | 0.21 ± 0.08 | 0.64 ± 0.15 | 0.82 ± 0.01 | 0.62 ± 0.05 |
| | RandomTrim | 0.02 ± 0.00 | 0.28 ± 0.06 | 0.72 ± 0.04 | 0.34 ± 0.03 | 0.19 ± 0.05 | 0.59 ± 0.14 | 0.82 ± 0.04 | 0.65 ± 0.04 |
| | Abstain (IDK) (Cheng et al., 2024) | 0.13 ± 0.00 | 0.13 ± 0.04 | 0.67 ± 0.04 | 0.02 ± 0.01 | 0.19 ± 0.02 | 0.38 ± 0.11 | 0.79 ± 0.04 | 0.08 ± 0.03 |
| | FactTune (Tian et al., 2023a) | 0.08 ± 0.00 | - | - | - | 0.31 ± 0.05 | - | - | - |
| | HALT (ours) | 0.09 ± 0.00 | **0.41** ± 0.27 | **0.74** ± 0.02 | **0.44** ± 0.06 | **0.57** ± 0.03 | **0.79** ± 0.60 | **0.86** ± 0.03 | **0.85** ± 0.08 |

Table 4: The table shows the average number of fragments per response for answers generated by each pretrained model using in-context learning. The 'Ground Truth' row represents the ground truth answer, with each category showing 100% correctness. Among the models, Llama-3-70B has the highest relative correctness, followed by Llama-3-8B and Gemma-2-9B, with Mistral-7B-v0.3 performing the lowest.

| Pretrained Model | Wikipedia | | Math | | Medical Q. | | Coding | | % corr. avg |
|---|---|---|---|---|---|---|---|---|---|
| | # given | % corr. | # given | % corr. | # given | % corr. | # given | % corr. | |
| Ground Truth | 9.8 ± 8.0 | 100.0 | 4.5 ± 3.4 | 100.0 | 5.1 ± 1.2 | 100.0 | 15.0 ± 12.7 | 100.0 | 100.0 |
| Llama-3-8B | 11.9 ± 11.2 | 12.9 | 3.3 ± 3.2 | 30.1 | 5.4 ± 1.5 | 66.9 | 9.3 ± 11.7 | 42.2 | 38.0 |
| Llama-3-70B | 9.6 ± 10.3 | 30.9 | 5.5 ± 3.0 | 54.1 | 5.2 ± 1.3 | 83.8 | 7.8 ± 9.0 | 58.8 | 56.9 |
| Gemma-2-9B | 9.6 ± 8.0 | 6.4 | 4.1 ± 3.3 | 28.6 | 5.3 ± 1.9 | 71.8 | 12.4 ± 10.2 | 30.2 | 34.3 |
| Mistral-7B-v0.3 | 9.6 ± 7.9 | 6.3 | 3.7 ± 3.1 | 17.1 | 5.2 ± 1.8 | 60.8 | 11.7 ± 9.1 | 27.9 | 28.0 |

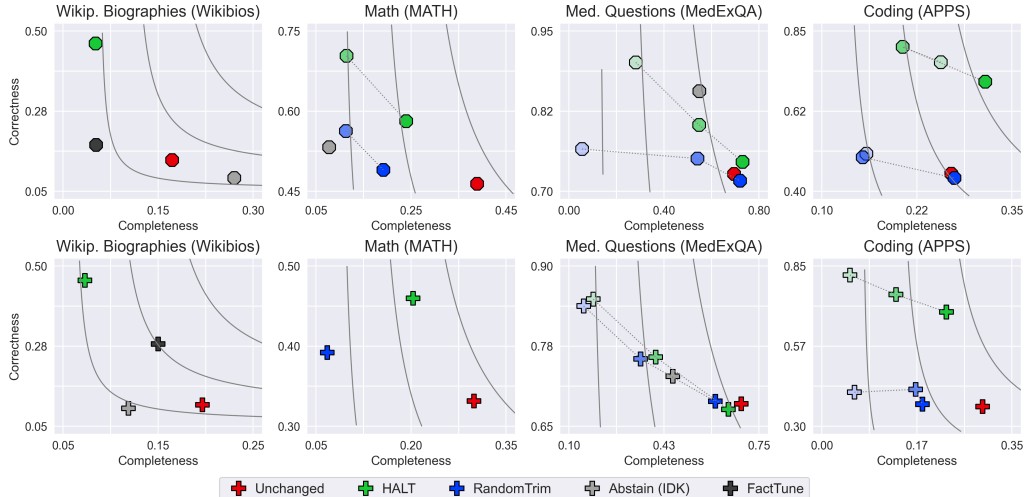

Figure 7: Response correctness (y-axis) and completeness (x-axis) for Gemma2-9B (top) and Mistral-7B (bottom) when finetuned with different methods (desired number of fragments is $n_{all}$). F1 Score is constant along curved grey lines, and highest in the top right corner. For HALT and Randomtrim, results are shown with different trade-offs between completeness and correctness, where lighter colors indicate tuning for higher correctness. We omit results with less than 5% completeness.

| Dataset | HALT Wins | Base Wins | Ties | Win Rate |
|---------|-----------|-----------|------|----------|
| MATH | 49 | 59 | 42 | 45.4% |
| appsintro | 46 | 64 | 39 | 41.8% |
| medex | 54 | 65 | 9 | 45.4% |
| wikibios_v2 | 97 | 50 | 3 | 66.0% |
| **Overall** | **246** | **238** | **93** | **50.8%** |

Table 5: AlpacaEval-style evaluation comparing HALT ($\alpha = 0.6$) against base Llama3-70B across 578 prompts. Win rate excludes ties. GPT-4o-mini served as judge.

## A.3 ALPACAEVAL GENERAL INSTRUCTION-FOLLOWING RESULTS

---

**Example prompt to Llama3-405B evaluator to determine correctness of a fragment that is part of a response composed of independent fragments.**

****Context:****
{{FULL WIKIPEDIA ARTICLE OF BARACK OBAMA}}
****Statement:****
Barack Obama was born on May 12, 1960.
****Instruction:****
Is the above Statement True or False according to the given Context? Reply with either 'True' or 'False.'

---

Figure 8: Example prompt to Evaluator for assessing a fragment from a Wikipedia-style response.

---

**Example prompt to Llama3-405B evaluator to determine correctness of a response composed of causally dependent fragments.**

****Math Question:****
A triangle has a base of 8 units and a height of 5 units. If the triangle is part of a rectangle with the same base and height, what fraction of the rectangle's area does the triangle occupy?
****Ground truth Response for Math Question:****
The area of a triangle is calculated using the formula: $\text{Area} = \frac{1}{2} \times \text{base} \times \text{height}$.

Substituting the given values: $\text{Area} = \frac{1}{2} \times 8 \times 5 = 20$ square units.

The area of the rectangle is calculated using the formula: $\text{Area} = \text{base} \times \text{height}$. Substituting the given values: $\text{Area} = 8 \times 5 = 40$ square units.

The triangle occupies a fraction of the rectangle's area given by: $\frac{\text{Triangle's area}}{\text{Rectangle's area}} = \frac{20}{40} = \frac{1}{2}$.

Thus, the triangle occupies $\frac{1}{2}$ of the rectangle's area.
****Step-wise Response given by student:****
Line 1: The area of a triangle is calculated using the formula: $\text{Area} = \frac{1}{2} \times \text{base} \times \text{height}$.

Line 2: Substituting the given values: $\text{Area} = \frac{1}{2} \times 8 \times 5 = 20$ square units.

Line 3: The area of the rectangle is $8 \times 5 = 40$ square units.
Line 4: The fraction of the rectangle's area that the triangle occupies is 20 square units.
****Instruction:****
You are given a Math question above, as well as a Ground truth Response to it. Determine until which line the step-wise response given by the student is correct. Reply with the number of the last correct line. If the first step is already wrong, reply with zero. Otherwise, reply with the number of the last correct line, i.e., with an integer between 0 and the number of lines in the response. Reply with 'Line X' only, where X is the number of the last correct line.

---

Figure 9: Example prompt to Evaluator for assessing a MATH response.

## B.2 EXAMPLE HALT RESPONSES FOR DIFFERENT TRADE OFFS BETWEEN RESPONSE CORRECTNESS AND RESPONSE COMPLETENESS

### B.2.1 EXAMPLE HALT FINETUNING RESPONSES FOR THE WIKIBIOS AND MATH DATASETS AND FOR DIFFERENT THRESHOLDS

We show examples of HALT responses for different trade-offs between response completenes and response correctness in Figures 10 and 11.

We note that responses for the Wikipedia biography dataset are segmented into individual atomic statements. This segmentation results fromt he application of the fact-extraction LLM (Song et al., 2024), but can reverted by prompting the LLM to recompose the response into a single text at the end of the generation. We omitted this recomposition as it complicates the assessment of individual statements.

**Example for $\alpha = 0.4$**

**\*\*\*\* Question:\*\*\*\***

Tell me about Daniel the Stylite

Respond with no more than 500 tokens. Only list facts about the person that you are certain about. Separate all facts by '\n'. If unsure how to continue at any point during the response, write 'Unsure from here.' and terminate your response.

**\*\*\*\* Response:\*\*\*\***
Saint Daniel the Stylite died on 11 December 493.

---

**Example for $\alpha = 0.6$**

**\*\*\*\* Question:\*\*\*\***

Tell me about Daniel the Stylite

Respond with no more than 500 tokens. Only list facts about the person that you are certain about. Separate all facts by '\n'. If unsure how to continue at any point during the response, write 'Unsure from here.' and terminate your response.

**\*\*\*\* Response:\*\*\*\***
Saint Daniel the Stylite was born around 409.

Saint Daniel the Stylite died on 11 December 493.

Saint Daniel the Stylite was a Christian saint.

---

**Example for $\alpha = 0.8$**

**\*\*\*\* Question:\*\*\*\***

Tell me about Daniel the Stylite

Respond with no more than 500 tokens. Only list facts about the person that you are certain about. Separate all facts by '\n'. If unsure how to continue at any point during the response, write 'Unsure from here.' and terminate your response.

**\*\*\*\* Response:\*\*\*\***
Saint Daniel the Stylite was born around 409.

Saint Daniel the Stylite died on 11 December 493.

Saint Daniel the Stylite was a Christian saint.

Saint Daniel the Stylite is commemorated on 11 December in the Eastern Orthodox Church.

Figure 10: Example HALT responses for different treshold parameters $\alpha$ for the Wikipedia dataset. Finetuning on the left response (lowest $\alpha$) results in a conservative LLM providing responses of higher correctness and lower completeness, while finetuning on right response results in a eagerly responding LLM.

**Finetuning pair for $\alpha = 0.4$**

**\*\*\*\* Question:\*\*\*\***

A triangle has a base of 8 units and a height of 5 units. If the triangle is part of a rectangle with the same base and height, what fraction of the rectangle's area does the triangle occupy?

**\*\*\*\* Response:\*\*\*\***
The area of a triangle is calculated using the formula: Area $= \frac{1}{2} \times$ base $\times$ height. Substituting the given values: Area $= \frac{1}{2} \times 8 \times 5 =$ 20 square units.
Unsure from here.

**Finetuning pair for $\alpha = 0.6$**

**\*\*\*\* Question:\*\*\*\***

A triangle has a base of 8 units and a height of 5 units. If the triangle is part of a rectangle with the same base and height, what fraction of the rectangle's area does the triangle occupy?

**\*\*\*\* Response:\*\*\*\***
The area of a triangle is calculated using the formula: Area $= \frac{1}{2} \times$ base $\times$ height. Substituting the given values: Area $= \frac{1}{2} \times 8 \times 5 =$ 20 square units.
The area of the rectangle is calculated using the formula: Area $=$ base $\times$ height. Substituting the given values: Area $= 8 \times 5 = 40$ square units.
Unsure from here.

**Finetuning pair for $\alpha = 0.8$**

**\*\*\*\* Question:\*\*\*\***

A triangle has a base of 8 units and a height of 5 units. If the triangle is part of a rectangle with the same base and height, what fraction of the rectangle's area does the triangle occupy?

**\*\*\*\* Response:\*\*\*\***
The area of a triangle is calculated using the formula: Area $= \frac{1}{2} \times$ base $\times$ height. Substituting the given values: Area $= \frac{1}{2} \times 8 \times 5 =$ 20 square units.
The area of the rectangle is calculated using the formula: Area $=$ base $\times$ height. Substituting the given values: Area $= 8 \times 5 = 40$ square units.
The triangle occupies a fraction of the rectangle's area given by: $\frac{\text{Triangle's area}}{\text{Rectangle's area}} = \frac{20}{40} = \frac{1}{2}$.
Thus, the triangle occupies $\frac{1}{2}$ of the rectangle's area.

Figure 11: Example HALT responses for different treshold parameters $\alpha$ for the MATH dataset. The left response (lowest $\alpha$) results in a conservative LLM providing responses of higher correctness and lower completeness, while finetuning on the right response results in a eagerly responding LLM.

