# OpenReview forum: "High Accuracy, Less Talk (HALT): Reliable LLMs through Capability-Aligned Finetuning"
_ICLR.cc/2026/Conference — ICLR 2026 Poster_

### Official Review · Reviewer_5Yi7 · 2025-10-28

**Soundness:** 3
**Presentation:** 4
**Contribution:** 3
**Rating:** 10
**Confidence:** 4

**Summary:**

The authors are addressing the issue of LLMs generating responses they are not confident in which can lead to hallucinations. They only want the LLM to generate responses it is confident in otherwise it'll stop generating certain responses at will output "Unsure".

The authors propose a post-training method called High Accuracy, Less Talk (HALT). This method involves prompting the LLM to generate responses for certain domains (wiki, math reasoning, etc). They then break these responses down into fragments and use an evaluator (separate LLM) to determine which fragments are correct / incorrect. They then create a new dataset where the responses are only the correct responses.

After finetuning the authors show that their model improves correctness on all their testsets along with maintaining a reasonable response completeness.

**Strengths:**

1) The method laid out is clearly explained and takes a principled approach. I think the different ways to assess fragments based off the domain is good and it is good that the authors are acknowledging future work would need to involve dependency graphs.

2) The baselines provided seem to be good to compare their method against.

3) The study in the section "Finetuning on Few-Shot Prompted Responses is comparable to Finetuning on Ground Truth
Responses" was important to run to trust this method. Even though it's been shown in previous that LLMs do not acquire novel capabilities during finetuning since the few-shot prompting was a key component of the method was good to show that results wont diminish much.

**Weaknesses:**

1) One thing that is not clear is how does response completeness affect user experience? Obviously we want correct responses for the user but is a completeness score of 51% low? What's the best tradeoff?

2) There's some lack of discussion. In Figure 4 why is there a sharp dropoff for the MATH dataset but not for Wikibios. I might be missing something but it would be good to have this clarified.

**Questions:**

Questions

1) Why are you using RandomTrim as a baseline? It's not clear what purpose it serves as a baseline.

2) Since you are sampling responses when creating the finetuning testset how often are the responses similar to each other? Does sampling help then if you are not getting diverse responses?

3) Have you considered using the negative fragments to move the model's response away from those?

Suggestions / Typos

1) I think you meant to say APPS in line 334

2) I suggest putting figures closer to where you mention them like Figure 3

---

> ### Author Response · Authors · 2025-11-22
>
> Thank you for assessing our work. We are pleased to hear that you find our approach "principled" and that our few-shot prompting analysis promotes "trust" in our method.
>
>
>
> > One thing that is not clear is how does response completeness affect user experience? Obviously we want correct responses for the user but is a completeness score of 51% low? What's the best tradeoff?
>
>
>
> HALT targets safety-critical domains (medicine, law, software development) where accuracy outweighs completeness. For use cases requiring complete responses, Add-On HALT (Section 5.3.3) annotates uncertain fragments as "Uncertain" instead of omitting them, maintaining completeness while reliably informing users which fragments are likely incorrect.
>
>
>
> > There's some lack of discussion. In Figure 4 why is there a sharp dropoff for the MATH dataset but not for Wikibios.
>
>
>
> This reflects the task structures: MATH uses causally dependent fragments (sequential reasoning) where one error invalidates subsequent steps, creating a sharp correctness threshold. WikiBios uses independent fragments (factual statements) that can be selectively included, enabling more gradual trade off.
>
>
>
> > Why are you using RandomTrim as a baseline? It's not clear what purpose it serves as a baseline.
>
>
>
> RandomTrim serves as a control baseline to isolate the value of capability-aware fragment removal. It randomly removes the last n fragments (n sampled from a Poisson distribution matching HALT's average response length) instead of using HALT's capability assessment. By controlling for response length while randomizing which fragments are removed, RandomTrim demonstrates that HALT's performance gains stem from intelligently selecting which fragments to keep based on model capability (not merely from producing shorter responses).
>
>
>
> > Since you are sampling responses when creating the finetuning testset how often are the responses similar to each other? Does sampling help then if you are not getting diverse responses?
>
>
>
> Our best-of-5 sampling approach provides sufficient diversity to estimate capability across different correctness levels. Empirically, we observed natural variance in fragment correctness across samples, particularly for questions where models have partial knowledge. When higher diversity is needed (e.g., for rare edge cases), standard techniques like temperature scaling or top-k sampling [9] can increase variation. The key insight is that HALT only requires enough diversity to distinguish between responses at different capability levels (defined by α), not to explore the entire response space.
>
>
>
> > Have you considered using the negative fragments to move the model's response away from those?
>
>
>
> We have not considered this approach but agree that a DPO-style method training on pairs could be an interesting direction for future work. However, such approaches require careful selection of negative examples to avoid reward hacking or overfitting to specific error patterns, whereas HALT's current approach of training only on capability-aligned positive examples is simpler and avoids these potential pitfalls.
>
>
>
> > I think you meant to say APPS in line 334. I suggest putting figures closer to where you mention them like Figure 3.
>
>
>
> Thank you for the feedback. We have corrected the naming inconsistency (changed "Wikibios" to "APPS") and moved Figures closer to where mentioned in the text.
>
>
> **References:**
>
> - [9] Brown et al., "Language Models are Few-Shot Learners", NeurIPS 2020

---

### Official Review · Reviewer_UdNm · 2025-10-29

**Soundness:** 3
**Presentation:** 2
**Contribution:** 2
**Rating:** 4
**Confidence:** 4

**Summary:**

HALT fine-tunes large language models to answer only when confident and to abstain when uncertain, outputting "Unsure from here". It trains models only on content they can reliably generate, aligning outputs with internal confidence rather than forcing full responses. HALT decomposes answers into factual fragments, verifies each with an evaluator LLM, and keeps only correct ones. Without adding inference cost, it improves factual accuracy and balances precision and recall. Across LLaMA3, Gemma2, and Mistral models, HALT outperforms standard finetuning, producing more reliable and self-aware responses.

**Strengths:**

1. The core idea of this paper is intuitive and clearly presented.
2. The "Unsure from here" mechanism makes the model’s uncertainty explicitly interpretable, improving user trust and enabling controllable trade-offs between correctness and completeness.

**Weaknesses:**

1. The method depends on an additional evaluator model to assess fragment correctness, which may introduce bias or inconsistency depending on the evaluator’s quality and alignment.
2. The method also relies on fragmentation, which feels somewhat heuristic to me.
3. The training process is complex, requiring additional time and computational cost.

**Questions:**

1. What does the statement f in line 147 refer to? I assume it corresponds to the "fragment" mentioned in the Introduction section, but please clarify this.
2. Why do you sample four prompt–response pairs and concatenate them with the target prompt?
3. Using Llama3-405B as the evaluator or for fragmentation appears quite costly. How sensitive is the proposed method to the evaluator model’s size and capability?

---

> ### Author Response · Authors · 2025-11-22
>
> Thank you for the review of our work. We are pleased to hear you find that our work enables "improving user trust and enabling controllable trade-offs between correctness and completeness".
>
>
>
> > The method depends on an additional evaluator model to assess fragment correctness, which may introduce bias or inconsistency depending on the evaluator's quality and alignment.
>
>
>
> While LLM evaluators have known limitations [4, 5], several factors ensure reliable evaluation: (1) we empirically validated our evaluator against 300 human-annotated responses per dataset (Table 2), achieving 0.27-1.14 fragment error rates; (2) our evaluator is information-grounded, assessing claims against provided evidence (Wikipedia articles, ground-truth solutions) rather than parametric knowledge; (3) systematic biases affect HALT and baselines equally, preserving comparative validity. Future work could further reduce bias through ensemble evaluation with multiple evaluator models or expanded human validation studies.
>
>
>
> > The method also relies on fragmentation, which feels somewhat heuristic.
>
>
>
> HALT's fragmentation is grounded in established methods for extracting atomic, verifiable statements from complex text [6, 7]. Correctness can only be meaningfully assessed at the atomic level—complete responses contain multiple claims that may have varying factuality, and fragment-level evaluation enables precise identification of which specific claims are correct.
>
>
>
> > The training process is complex, requiring additional time and computational cost.
>
>
>
> While HALT adds preprocessing steps (few-shot sampling, fragmentation, evaluation), the computational overhead is modest and one-time. The evaluator runs once per training sample during dataset preparation (not inference), and fragmentation is deterministic. Most importantly, HALT requires no additional inference cost compared to standard finetuning—the deployed model has identical architecture and runtime.
>
>
>
> > What does the statement f in line 147 refer to?
>
>
>
> Thank you for pointing this out. The notation `f` refers to individual fragments, as defined in Section 4 where we denote a response split into fragments as y^pt = (f_1, ..., f_k). We have clarified this in the revised manuscript by explicitly defining the fragment notation when first introduced in the evaluator paragraph.
>
>
>
> > Why do you sample four prompt–response pairs and concatenate them with the target prompt?
>
>
>
> We sample four prompt-response pairs from the training dataset and concatenate them with the target prompt to provide in-context learning examples [9] to the pretrained model. This few-shot prompting approach helps the model understand the expected response format and style.
>
>
>
> > Using Llama3-405B as the evaluator or for fragmentation appears quite costly. How sensitive is the proposed method to the evaluator model's size and capability?
>
>
>
> The evaluator runs only once per training sample during dataset preparation, not inference, amortizing costs. Crucially, our evaluator is information-grounded—conditioned on evidence (Wikipedia articles, ground-truth solutions)—reducing capability requirements versus parametric evaluation. We expect mid-size models suffice for this context-conditioned task. Since high-quality post-training may require only ~1,000 samples [8], even expensive evaluation remains practical.
>
> **Ask to Reviewer:**
> We appreciate your recognition that our work "enables improving user trust and controllable trade-offs between correctness and completeness." Given our responses addressing the evaluator quality, fragmentation grounding, and practical cost considerations, we respectfully ask you to consider raising your score. If any concerns remain that place our work below the acceptance threshold, we would be grateful if you could point them out so we can address them.
>
> **References:**
> - [4] Panickssery et al., "LLM Evaluators Recognize and Favor Their Own Generations", NeurIPS 2024
> - [5] Fu et al., "Are Large Language Models Reliable Judges? A Study on the Factuality Evaluation Capabilities of LLMs", GEM Workshop 2023
> - [6] Song et al., "VERISCORE: Evaluating the factuality of verifiable claims in long-form text generation", 2024
> - [7] Min et al., "FactScore: Fine-grained atomic evaluation of factual precision in long form text generation", 2023
> - [8] Zhou et al., "LIMA: Less is More for Alignment", NeurIPS 2024
> - [9] Brown et al., "Language Models are Few-Shot Learners", NeurIPS 2020

---

### Official Review · Reviewer_i3Pq · 2025-11-01

**Soundness:** 3
**Presentation:** 2
**Contribution:** 2
**Rating:** 4
**Confidence:** 4

**Summary:**

The authors propose HALT, a method to train language models to abstain from providing certain fragments of their long-form responses when they are uncertain. Given an LM, the approach obtains several samples for each training input, splits each sample into fragments and fact-checks them, and finally replaces incorrect fragments with abstention statements. By collecting samples with more or fewer correct statements into different supervised finetuning splits, HALT is able to navigate the correctness-completeness tradeoff for long-form generation, unlike prior work. Experiments on four datasets across general-domain factuality, math, etc., demonstrate HALT's navigation of this tradeoff.

**Strengths:**

- The paper is generally well-written and clear.
- The motivation is strong; abstention and uncertainty quantification at claim-level are an important approach to decrease LM hallucination while retaining usefulness.
- The method is simple and clever, in particular the idea to use multiple samples given an input along with their number of correct claims to control the correctness-completeness tradeoff. I suspect that this work could emerge as a straightforward, effective baseline for training LMs to abstain in their long-form generations.
- The experiments are reasonably thorough (four datasets in different domains, evaluation of multi-task transfer, and extension to uncertainty quantification).

**Weaknesses:**

- The authors should better contextualize with respect to prior work on finetuning language models to express their verbalized uncertainty [1] [2]. That setting is in some sense more challenging than the final experiment in the present paper, which marks fragments as uncertain instead of omitting them.
- A number of heuristic choices are made which are not clearly ablated. For example, I would expect that many non-mathematical tasks have some causal dependency in consecutive sentences, and therefore simply replacing an arbitrary subset of fragments with an abstention phrase will harm coherence. Can you evaluate HALT on some "guardrail" instruction-following tasks which would test how coherence and helpfulness are affected by your finetuning, e.g., AlpacaEval or similar?

[1] Band et al., Linguistic Calibration of Long-Form Generations. ICML 2024. https://arxiv.org/abs/2404.00474
[2] Stengel-Eskin et al., LACIE: Listener-Aware Finetuning for Confidence Calibration in Large Language Models. NeurIPS 2024. https://arxiv.org/abs/2405.21028

**Questions:**

See above for the key feedback on better contextualization with respect to previous "LM finetuning for calibration" works, and evaluating coherence with a guardrail task.

---

> ### Author Response · Authors · 2025-11-22
>
> Thank you for the thorough assessment of our work. We are pleased to hear you consider it an "important approach to decrease LM hallucination" and that you find that our work "could emerge as an effective baseline for training LMs to abstain".
>
> > The authors should better contextualize with respect to prior work on finetuning language models to express their verbalized uncertainty [1] [2]. That setting is in some sense more challenging than the final experiment in the present paper, which marks fragments as uncertain instead of omitting them.
>
> Thank you for pointing out these related works. We have added them to our related work (Section 3).
>
> **Band et al. [1]** train models to express explicit confidence scores (e.g., "X is true (confidence: 30%)"), while **Stengel-Eskin et al. [2]** use listener-aware training to calibrate hedging language. Both preserve complete responses with uncertainty markers, requiring users to interpret confidence levels.
>
> HALT  -- rather than expressing uncertainty while keeping potentially incorrect content -- selectively omits uncertain fragments entirely. We believe that all approaches have their merits but that HALTs approach of omitting potentially incorrect segments can be especially useful in high-stakes domains, where any potenitally incorrect inforamtion could bias users.
> Additionally, HALT enables controlling the correctness-completeness tradeoff through the α parameter.
>
> > A number of heuristic choices are made which are not clearly ablated. For example, I would expect that many non-mathematical tasks have some causal dependency in consecutive sentences, and therefore simply replacing an arbitrary subset of fragments with an abstention phrase will harm coherence. Can you evaluate HALT on some "guardrail" instruction-following tasks which would test how coherence and helpfulness are affected by your finetuning, e.g., AlpacaEval or similar?
>
> We have conducted an AlpacaEval-style evaluation [3] to assess whether HALT maintains coherence and instruction-following capabilities, which we have added to Section 5.2 of the revised manuscript. We compared the Llama3-70B HALT model (trained with α = 0.6) against the base Llama3-70B model across 578 prompts spanning four datasets: MATH (150), appsintro (150), medex (128), and wikibios_v2 (150). Using GPT-5 as the judge, we measured how often each model's responses were preferred.
>
> Results:
> | Dataset      | HALT Wins | Base Wins | Win Rate |
> |--------------|-----------|-----------|----------|
> | MATH         | 49        | 59        | 45.4%    |
> | appsintro    | 46        | 64        | 41.8%    |
> | medex        | 54        | 65        | 45.4%    |
> | wikibios_v2  | 97        | 50        | 66.0%    |
> | **Overall**  | **246**   | **238**   | **50.8%** |
>
> HALT achieves near-parity with the base model overall (50.8% win rate), demonstrating that selective abstention does not significantly harm general instruction-following capabilities. Notably, HALT outperforms the base model on wikibios_v2 (66.0% win rate), where factual accuracy is critical—precisely the use case HALT is designed for. On math and general-domain tasks (MATH, appsintro, medex), HALT shows modest degradation (41.8-45.4%), which is expected since the judge may penalize abstentions on tasks where completeness is valued over guaranteed correctness, with the judge potentially being unable to perfectly judge correctness.
>
> **Ask to Reviewer:**
> Given the additional AlpacaEval results demonstrating that HALT maintains instruction-following capabilities, combined with your overall positive assessment, we respectfully ask you to consider raising your score. If any concerns remain that place our work below the acceptance threshold, we would be grateful if you could point them out so we can address them.
>
> **References:**
> - [1] Band et al., "Linguistic Calibration of Long-Form Generations", ICML 2024
> - [2] Stengel-Eskin et al., "LACIE: Listener-Aware Finetuning for Confidence Calibration in Large Language Models", NeurIPS 2024
> - [3] Li et al., "AlpacaEval: An Automatic Evaluator of Instruction-following Models", 2023

---

> > ### Comment · Reviewer_i3Pq · 2025-11-22
> >
> > I appreciate the authors' efforts to expand related work and include an evaluation demonstrating that HALT generally preserves coherence. I see HALT as a clean baseline for abstention in LLMs, which is a valuable contribution, and therefore raise my score to the acceptance regime.

---

### Author Response · Authors · 2025-11-22
**Response to Reviewers**

Dear reviewers,

Thank you for the detailed feedback. We're pleased that you found our work "important" (R1), "principled" (R3), and effective at "improving user trust and enabling controllable trade-offs" (R2).

Based on your feedback, we made changes to the manuscript. Most importantly:

1. **AlpacaEval evaluation** (Section 5.2): HALT achieves 50.8% win rate overall vs. the base model, with strong performance on factual tasks (66.0% on WikiBios), showing that selective abstention doesn't harm instruction-following.

2. **Related work on verbalized uncertainty** (Section 3): We added discussion of Band et al. and Stengel-Eskin et al., explaining how HALT's selective omission differs from approaches that express uncertainty while keeping potentially incorrect content.

3. **Technical clarifications**: We clarified fragment notation, grounded our fragmentation in VeriScore/FactScore, and explained computational costs, e.g. one-time preprocessing, zero inference overhead.

Please feel free to raise any concerns or ask any questions.

Kind regards,
The authors

---

### Meta-Review · Area_Chair_F6AW · 2025-12-28

**Summary:**

The paper proposes HALT, a post-training method that finetunes LLMs to generate content only when confident, by identifying and removing or masking incorrect fragments from pretrained responses, resulting in models that significantly improve correctness (e.g., 51% → 87% in Llama3-70B) while allowing a tunable trade-off between response completeness and reliability across multiple domains.

The common concerns across reviewers focused on the heuristic nature of HALT, potential impacts on coherence when replacing fragments with abstention phrases, reliance on an external evaluator, and the computational cost of training. After the rebuttal, the authors have satisfactorily addressed all these concerns, providing clarifications, comparison to related works, and AlpacaEval evaluation. The paper is well-written and easy to understand, the method design is well-motivated, and this training approach will meaningfully help mitigate hallucinations in LLMs, making it an impactful contribution to the field.

Overall, AC recommends acceptance.

**Reviewer Concerns:**

The initial concerns from all the reviewers are summarized below:

Reviewer i3Pq: The reviewer is concerned that the paper lacks contextualization with prior work on finetuning LLMs to express uncertainty. Several heuristic choices, such as fragment replacement with abstention phrases, are not ablated, and may harm coherence in tasks with causal dependencies.

Reviewer UdNm: The method’s dependence on an external evaluator to assess fragment correctness may introduce bias or inconsistency. Fragmentation itself is viewed as heuristic, and the training process is complex and computationally costly. Questions include clarifying what “f” refers to, why multiple prompt–response pairs are concatenated, and how sensitive HALT is to the size and capability of the evaluator model.

Reviewer 5Yi7: The reviewer questions the impact of response completeness on user experience and the optimal tradeoff between correctness and completeness. They request clarification on the sharp dropoff for the MATH dataset in Figure 4 and the rationale for using RandomTrim as a baseline. Additional questions include diversity of sampled responses and whether negative fragments could be used to guide the model away from incorrect responses.

The authors have done an excellent job providing clarifications, and all the initial concerns raised about the paper have been satisfactorily addressed.

**Reviewer Scores:**

Reviewer i3Pq and Reviewer UdNm initially gave scores of 4, while Reviewer 5Yi7 gave a score of 10. After reading the rebuttal, the AC is confident that the two reviewers who initially scored 4 will raise their scores, as their questions and concerns have been thoroughly clarified and addressed.

---

### Decision · Program_Chairs · 2026-01-26

Accept (Poster)